# Improving Underwater Continuous-Variable Measurement-Device-Independent Quantum Key Distribution via Zero-Photon Catalysis

**DOI:** 10.3390/e22050571

**Published:** 2020-05-19

**Authors:** Yuang Wang, Shanhua Zou, Yun Mao, Ying Guo

**Affiliations:** 1School of Automation, Central South University, Changsha 410083, China; wya1759991046@gmail.com (Y.W.); maocsu@sina.com (Y.M.); 2School of Internet of Things Engineering, Wuxi Taihu University, Wuxi 214064, China; 3State Key Laboratory of Advanced Optical Communication Systems and Networks, Shanghai Jiao Tong University, Shanghai 200240, China

**Keywords:** continuous-variable quantum key distribution, measurement device independent, zero-photon catalysis, underwater channel

## Abstract

Underwater quantumkey distribution (QKD) is tough but important formodern underwater communications in an insecure environment. It can guarantee secure underwater communication between submarines and enhance safety for critical network nodes. To enhance the performance of continuous-variable quantumkey distribution (CVQKD) underwater in terms ofmaximal transmission distance and secret key rate as well, we adopt measurement-device-independent (MDI) quantum key distribution with the zero-photon catalysis (ZPC) performed at the emitter of one side, which is the ZPC-based MDI-CVQKD. Numerical simulation shows that the ZPC-involved scheme, which is a Gaussian operation in essence, works better than the single photon subtraction (SPS)-involved scheme in the extreme asymmetric case. We find that the transmission of the ZPC-involved scheme is longer than that of the SPS-involved scheme. In addition, we consider the effects of temperature, salinity and solar elevation angle on the system performance in pure seawater. The maximal transmission distance decreases with the increase of temperature and the decrease of sunlight elevation angle, while it changes little over a broad range of salinity

## 1. Introduction

Quantum key distribution (QKD) [1,2,3] is a key part of quantum communications. There are two categories of protocols, that is, the discrete-variable (DV) QKD protocol [4,5] and the continuous variable (CV) QKD protocol [6,7,8]. DVQKD, which was proposed in 1984 with the proposal of Bennett-Brassard 1984 (BB84) [9], codes on different states of a single photon to convey information. Currently, it has gotten fully developed and has been experimented in free space, optical fiber, and so forth. However, DVQKD can be easily interfered by various factors such as background noise light and noise from components. Besides, because single-photon source is quite hard to realize even nowadays, people use attenuating laser sources for substitution, which could exert bad effects on secret key rate. Fortunately, two decades after BB84 was proposed, CVQKD was born, which was based on the continuity of quantum eigenstate and modulates information on continuous variable of quantum such as phase and amplitude for communications. Compared with DVQKD, CVQKD can automatically filter background noise light with simple light source at the same time. Subsequently, CVQKD is compatible with contemporary optical communication system, which makes it a hot topic in QKD realm quickly. Moreover, in terms of measurement devices, CVQKD relies on homodyne or heterodyne detectors, which are more efficient to achieve higher secret key rates than single-photon detectors. Of course, CVQKD is still imperfect. There exist disadvantages like short transmission distance, but these defects are being overcome by advancing technology.

Currently, there have been several CVQKD protocols in terms of system model, such as the point-to-point (PP) CVQKD and measurement-device-independent [10,11] (MDI) CVQKD [12]. PP-CVQKD, as literally interpreted, is conducted between two parties, Alice and Bob, directly. It is vulnerable to attacks aimed at detector imperfection. However, in MDI-CVQKD, Alice and Bob first prepare and transmit coherent states to the third party Charlie. Subsequently, Charlie interferes the received states to make Bell measurement and announces measurement results publicly. Finally, the secret key can be shared between Alice and Bob after post-processing. Compared with PP-CVQKD, MDI-CVQKD is born to solve the flaw of detector imperfection. It can resist side-channel attacks such as the local oscillator calibration attack [13], the wavelength attack [14], and the detector saturation attack [15].

At present, CVQKD is always conducted through free space and fiber channel, both of which are meaningful but challenging. Light transmission in air channel can be disturbed by natural environment like atmospheric turbulence [16,17,18], rain, fog, sunlight, and so forth. Fiber channel seems immune to external disturbance, but it is difficult to be wired up and could be easily destroyed. Underwater CVQKD may be more meaningful than air or fiber channel in a sense. Common QKD methods for two underwater vehicles nowadays are using periscopes and satellite link. However, these methods require underwater vehicles to rise to the sea surface. Fortunately, CVQKD can be feasibly implemented through underwater channel in practice, which provide a more convenient scheme for underwater vehicles to communicate safely. However, the realization of underwater CVQKD is more difficult considering attenuation caused by ocean current, molecular impact, microorganism, scattering, and so forth. These factors could exert adverse effects on entanglement between quantum, thus leading to short transmission distance. In what follows, we consider something different as the effects of temperature, salinity and sun elevation angle.

Recently, there have been several works for QKD underwater. For example, John proposed the underwater BB84 protocol using pairs of polarization entangled photons [19]. Bouchard suggested a high dimensional BB84 protocol with twisted photons in outdoor conditions [20]. After that Ruan proposed a method to estimate parameters to improve CVQKD performance [21]. However, the implementation of MDI-QKD underwater has been waiting for some researches to fill the gaps. Note that despite the absolute device security of MDI-QKD, its transmission distance is unsatisfactory, and thus it is difficult to be implemented in harsh environments like seawater. Fortunately, to lengthen the transmission distance, the non-Gaussian operations [22] like single photon subtraction (SPS) [23] and zero-photon catalysis (ZPC) [24] are the most commonly used means. One article has put forward a plan of operating single photon subtraction (SPS) in the fiber-based CVQKD [25]. In this paper, we dedicate to lengthen the transmission distance of underwater CVQKD via the Gaussian operations. Motivated by the characteristics of noiseless attenuation, we perform the zero-photon catalysis, which can keep the Gaussian behavior of photon to prolong the maximal transmission distance of the CVQKD system underwater with the achievable high secret key rate.

This paper is structured as follows. In Section 2, we propose the ZPC-based MDI-CVQKD for underwater secure communication. In Section 3, we show the performance improvement of the ZPC-based scheme by using numerical simulations. Finally, a conclusion is drawn in Section 4.

## 2. The ZPC-Based MDI-CVQKD Protocol

In this section, we suggest the ZPC-based MDI-CVQKD system through underwater channel. Due to the equivalence of prepare-and-measure (PM) scheme and entanglement-based (EB) scheme, we consider the EB ZPC-involved scheme to simplify the security proof of the underwater MDI-CVQKD system.

Figure 1 shows the schematic diagram of the EB ZPC-involved scheme. In this scheme, Alice in deep water aims to establish a secret channel with Bob in shallow water. Note that Alice and Bob may not locate in the same vertical area. For the convenience of demonstration, we suppose that Alice is vertically below Bob, and the transmission distance turns into depth. First, Alice and Bob prepare entanglement resource EPR1 and EPR2 with variances $V_{A}$ and $V_{B}$, respectively. Then, they keep modes $A_{1}$ and $B_{1}$, and send other modes $A_{2}$ and $B_{2}$ to an untrusted party Charlie through water channel. To simplify equipment, we assume that the ZPC operation is conducted by David on Alice’s side, which turns mode $A_{2}$ into mode $\tilde{A_{2}}$. After that, Charlie receives modes $\tilde{A_{2}}$ and $B_{2}$, and performs BSM (Bell state measurement)-based detection and announces measurement results $P_{C_{2}}$ and $X_{C_{1}}$ publicly through a classical channel. Ultimately, Bob modifies mode $B_{1}$ to mode $\tilde{B_{1}}$ through operation D(α), where D(α) is a displacement operation. In this way, Alice and Bob obtain two mode $A_{1}$, $\tilde{B_{1}}$ for heterodyne detection to get data $\left(X_{A},P_{A}\right)$ and $\left(X_{B},P_{B}\right)$, which can be used for estimation of channel parameter, coordinate information, and so forth. After series of post-processing, secret key will be achieved successfully.

As for the ZPC-involved data-processing shown in Figure 1 (a), vacuum state in auxiliary mode D is injected into an input port of beam splitter (BS) with transmittance *T*, which is detected at the corresponding output port of BS at the same time. That is exactly the ZPC operation. This process is usually represented by an equivalent operator given by
(1)$$\overset{\wedge }{O_{0}}\equiv Tr\left[B\left(T\right)\overset{\wedge }{\prod_{\mathrm{off}}}\right]={}_{D}\langle 0|B\left(T\right){|0\rangle }_{D},$$
where B(T) is the operator representing BS with transmittance *T* and can be described as
(2)$$B\left(T\right)=\exp\left[\sqrt{T}-1\right)\left({a_{2}}^{†}a_{2}+d^{†}d\right)+\left(d^{†}a_{2}-d{a_{2}}^{†}\right)\sqrt{1-T}],$$
and $\prod_{\mathrm{off}}^{\wedge }$ is the projection operator in photon detector(PD), which here is an on/off detector. Now we consider how the ZPC operation makes effect. State EPR1 is essentially a two-mode squeezed vacuum state, which can be expressed as
(3)$$\begin{array}{c} |EPR_{1}\rangle_{A_{1}A_{2}}=S_{2}\left(r\right)|0,0\rangle_{A_{1}A_{2}}\\ =\sqrt{1-\lambda^{2}}\sum_{l=0}^{\infty }\lambda^{l}|l,l\rangle_{A_{1}A_{2}}, \end{array}$$
where $\lambda =\sqrt{\left(V_{A}-1\right)\left(V_{A}+1\right)}$. After conducting the ZPC operation, this state turns into $|\psi \rangle_{A_{1}\tilde{A_{2}}}$, which can be described as
(4)$$|\psi \rangle_{A_{1}\tilde{A_{2}}}=\frac{{\hat{O}}_{0}}{\sqrt{P_{d}}}|EPR_{1}\rangle_{A_{1}A_{2}},$$
where $P_{d}=2/(1+T+(1-T)V_{A})$, standing for the success probability of the ZPC operation. Subsequently, the covariance matrix of $|\psi \rangle_{A_{1}\tilde{A_{2}}}$ can be calculated as
(5)$$V_{A_{1}\tilde{A_{2}}}=\left(\begin{array}{cc} x\prod & z\sigma_{z}\\ z\sigma_{z} & y\prod \end{array}\right),$$
where $\sigma_{z}=\mathrm{diag}\left(1,-1\right)$, $x=y=\left(2V_{A}-RV_{A}+R\right)/\left(1+T+RV_{A}\right)$, and $z=2\sqrt{T({V_{A}}^{2}-1)}/\left(1+T+RV_{A}\right)$. We note that the above-mentioned ZPC operation is actually a Gaussian operation in essence, which have an effect on the performance of the underwater CVQKD system.

## 3. Security Analysis

While demonstrating the effect of the ZPC-involved scheme on the underwater CVQKD system, we consider transmittance of seawater channel, which characterizes the transparency of seawater, thus affecting the ability of light transmission, which is shown in Appendix A. Subsequently, we show the performance improvement of the ZPC-based system.

### 3.1. Derivation of the Secret Key Rate

As shown in Figure 2, we have an equivalent point-to-point (PP) protocol of the underwater ZPC-based MDI-CVQKD. It should be noticed that the reasonableness of this equivalence has been proved [26]. Thus we use $T_{c}$ and $\varepsilon_{th}$ to represent the transmittance and excess noise of the PP CVQKD protocol given by
(6)$$T_{c}=g^{2}T_{A}/2,$$
and
(7)$$\varepsilon_{th}=T_{B}/T_{A}\left(\varepsilon_{B}-2\right)+\varepsilon_{A}+2/T_{A}.$$
Taking into account the noise caused by Charlie’s imperfect detection, the whole channel noise can be expressed as
(8)$$\chi_{tot}=1-T_{c}/T_{c}+\varepsilon_{th}+2\chi_{\mathrm{hom}}/T_{A},$$
with $\chi_{\mathrm{hom}}=\left(\nu_{el}+1-\eta \right)/\eta$, where $\nu_{el}$ stands for electronic noise and η stands for quantum efficiency. The transmittance $T_{A(B)}$ of seawater channel can be expressed as
(9)$$T_{A(B)}=e^{-\alpha \left(\lambda \right)D_{AC(BC)}},$$
where α(λ) means attenuation coefficient shown in Appendix A.

Different from non-Gaussian operation, after performing ZPC, the resulting state $|\psi \rangle_{A_{1}\tilde{A_{2}}}$ is still a Gaussian state, thus it is reasonable to derive the secret key rate directly from the conventional Gaussian CVQKD given by
(10)$$K=P_{d}\left\{\left(\beta I\left(A:B\right)\right)-\chi \left(B:E\right)\right\},$$
where β means the reverse-reconciliation efficiency, I(A:B) represents the mutual information between Alice and Bob, and χ(B:E) denotes the Holevo bound between Bob and Eve. Assuming $|\psi \rangle_{A_{1}\tilde{B_{1}}}$ denotes the state when $|\psi \rangle_{A_{1}\tilde{A_{2}}}$ passes through the channel in the equivalent PP CVQKD protocol, the covariance matrix of $|\psi \rangle_{A_{1}\tilde{B_{1}}}$ can be described as
(11)$$\begin{array}{c} V_{A_{1}\tilde{B_{1}}}=\left(\begin{array}{cc} X\prod & Z\sigma_{z}\\ Z\sigma_{z} & Y\prod \end{array}\right)\\ =\left(\begin{array}{cc} x\prod & \sqrt{T_{c}}z\sigma_{z}\\ \sqrt{T_{c}}z\sigma_{z} & T_{c}\left(x+\chi_{tot}\right)\prod \end{array}\right). \end{array}$$
Then, I(A:B) can be calculated as
(12)$$I\left(A:B\right)=\log_{2}\frac{\left(X+1)(Y+1\right)}{\left(X+1\right)\left(Y+1\right)-Z^{2}}.$$
To calculate χ(B:E), we assume Eve is aware of David’s existence and can purify the whole system $\rho_{A_{1}\tilde{B_{1}}ED}$. Based on this, χ(B:E) can be described as
(13)$$\begin{array}{c} \chi (B:E)=S(E)-S(E|B)\\ =\sum_{i=1}^{2}G(\frac{\lambda_{i}-1}{2})-G\left(\frac{\lambda_{3}-1}{2}\right), \end{array}$$
where $G\left(x\right)=\left(x+1\right)\log_{2}\left(x+1\right)-x\log_{2}x$, representing the von Neumann entropy, and $\lambda_{1,2}^{2}=\left(\Delta \pm \sqrt{\Delta^{2}-4\omega^{2}}\right)/2$ with $\omega \;=\;XY-Z^{2}$ and $\Delta \;=\;X^{2}+Y^{2}-2Z^{2}$.

### 3.2. Numerical Simulations

In the following, we show the performance improvement of the ZPC-based MDI-CVQKD in terms of the maximal transmission distance and the secret key rate as well, compared with the SPS-based MDI-CVQKD and the traditional MDI-CVQKD.

In numerical simulations of the secret key rate of the ZPC-based MDI-CVQKD, we set $D_{BC}=0$, which is the asymmetric case that achieves the longest transmission distance. Moreover, we take into account $\varepsilon_{A}=\varepsilon_{B}=0.01$, β=0.96, η=1, and $\nu_{el}=0$. First of all, we consider the influence of the tunable variance $V_{A}$ and $V_{B}$, where $V_{A}$ and $V_{B}$ are significant to system, as shown in Figure 3. For the simplicity, we set $V_{A}=V_{B}$. We find that the traditional scheme is sensitive to $V_{A}$($V_{B}$), whereas the SPS-based and ZPC-based schemes show the stable transmission depth even when $V_{A}$($V_{B}$) changes in a big range in Figure 3a. In addition, the secret key rate decreases fast with the increase of $V_{A}$($V_{B}$), as shown in Figure 3b. By contrast, the secret key rate of the other two schemes decrease slowly with the increase of $V_{A}$($V_{B}$). This result shows that the ZPC-based and SPS-based schemes have a more flexible application in the underwater CVQKD system.

Note that in practical system, the performance of CVQKD is related to the perfection of components. For example, the Faraday-mirror, which is used for adjusting the polarization angle of signal, is quite sensitive to the rotation angle. The rotation angle should be set as $45^{{}^{\circ}}$ accurately to make the polarization angles of signal and local oscillator orthogonal. However, in practice, the rotation angle could not be perfectly set, thus leading to the decrease of secret key rate, especially when transmittance *T* is small. Fortunately, increasing variance appropriately can provide us an efficient ploy to make up for the defects [27].

In Figure 4, we illustrate the performance of the related schemes in terms of the secret key rate and the maximal transmission depth under different variance. From Figure 4a, when variance $V_{A}$ ($V_{B}$) is small, both underwater ZPC-based and SPS-based schemes show no obvious advantages in terms of depth compared with the condition on land. For the SPS-based scheme, it reaches the longest depth at about 43 m, which is close to that of the traditional scheme. For the ZPC-based scheme, it has the longest transmission distance of 50 m. This phenomenon may be caused by the small transmittance in the sea. Due to the small transmittance of seawater, the secret key rate of all three schemes comes to zero fast, thus giving fewer chances for the SPS-based scheme and ZPC-based scheme to show distance advantages. However, In Figure 4b, it shows a different result. When variance $V_{A}$ ($V_{B}$) is increased, the longest distance of traditional scheme decreases to 30 m, while the performance of the SPS-based and ZPC-based schemes maintain stable. It seems that for the increased modulation variance the SPS-based and ZPC-based schemes show better performance than the traditional protocol, of which the ZPC operation works better. Moreover, it also shows that for the high modulation variance, the ZPC-based scheme is the best among the three schemes discussed above.

To show the advantages of the ZPC-based scheme over the SPS-based scheme, we plot the secret key rate as a function of transmittance (T) of beam splitter (BS) and depth. As shown in Figure 5, the ZPC-based scheme has apparent advantages in terms of both secret key rate and depth compared with the SPS-based scheme. Besides, from this figure, we can get the optimal transmittance (T) of both two schemes. We find that the optimal transmittance (T) is 0.75 for the ZPC-based scheme and 0.72 for the SPS-based scheme. This result proves that the ZPC operation does improve system performance and works better than the SPS operation.

Subsequently, we consider effects of factors of pure sea water on the ZPC-based MDI-CVQKD system. First of all, we consider the effects of temperature in Figure 6. It shows that the transmission depth changes by about 5 m when the temperature ranges from $0{\;}^{{}^{\circ}}$C to $40{\;}^{{}^{\circ}}$C. It seems that the colder the seawater means the better the performance. This characteristic is easily to be comprehended since colder seawater means weaker thermal movement of molecular, thus leading to weaker influence on the performance of the underwater CVQKD system. It should be noticed that this range of change is possible, considering differences in seasons, time in a day and geographical location.

Figure 7 shows the effects of sun elevation angle. Here we consider the influence that sunlight exerts on transmittance and omit the influence on the excess noise. The reason for this simplification is based on the assumption that the photon detector is ideal and not affected by background light. It is shown that depth lengthens by about 15 m when the sun elevation angle changes from $70{}^{{}^{\circ}}$ to $20{}^{{}^{\circ}}$. Therefore, we could deduce that the underwater CVQKD system has the best performance around midday and has the worst performance at dusk. This result is quite different from the situation of CVQKD in free space, transmittance of which has little relationship to background light while background noise is influenced profoundly by background solar light.

From simulation above, we can find that even if ZPC operation improves the performance of CV-MDI-QKD to some extent, our scheme is still constrained by transmission distance compared with conditions in fiber and open air, which is secure up to at least 100 km. However, its flexibility compared with fiber allows it to become the next generation of optical switch underwater. For example, it can be used as a non-contact optical switch to establish secure net for underwater vehicles. Besides, it can be applied to optical communication system for autonomous underwater robots [29] and remote underwater robot operation [30]. Moreover, the development of underwater wireless optical communication (UWOC) provides another chance for our scheme. Recently, Sun verified the operation of UWOC at tens of gigabits per second or close to a hundred meters of distance [31]. With the help of our proposed scheme, UWOC will be safer and more credible.

## 4. Conclusions

We have proposed a ZPC-involved scheme for strengthening the security of the underwater MDI-CVQKD system in terms of the secret key rate and the maximal transmission depth. This scheme aims to establish a potential underwater MDI-CVQKD channel between two underwater parties. We consider the influence that the ZPC operation exerts on the MDI-CVQKD system and derive the secret key rate. To make it more persuasive, we compare the ZPC-involved scheme with the SPS-involved and traditional schemes as well. Numerical simulations show that the ZPC-involved scheme has better performance, prolonging the transmission depth by about 5 m. We find that the ZPC-involved scheme shows better performance obviously when the tunable modulation variance is set high. Besides, we consider the possible factors influencing our proposed method. It is found that temperature has a relatively considerable impact on transmission depth while salinity is not an important factor in terms of the maximal transmission depth and the secret key rate. In addition, sun elevation angle influences the system performance to some extent as well, which implies that the performance of the underwater CVQKD system may be changeable with different time.

## Figures and Tables

**Figure 1 entropy-22-00571-f001:**
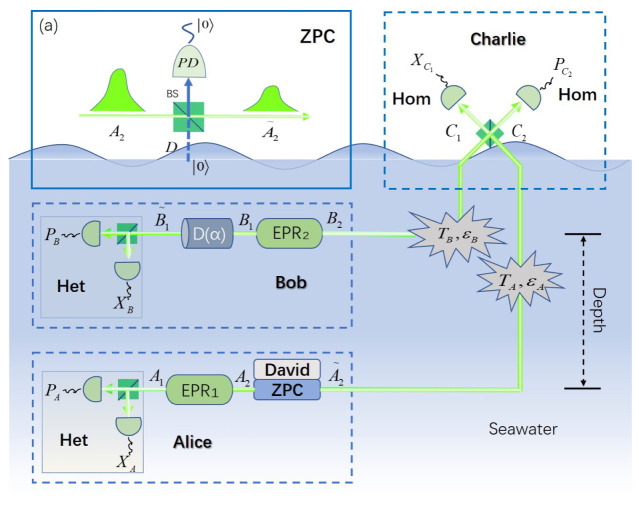
Schematic diagram of the zero-photon catalysis (ZPC) based measurement-device-independent-continuous-variable quantum key distribution (MDI-CVQKD) through underwater channel. Hom: homodyne detection, Het: heterodyne detection, PD: photon detector, BS: beam splitter.

**Figure 2 entropy-22-00571-f002:**
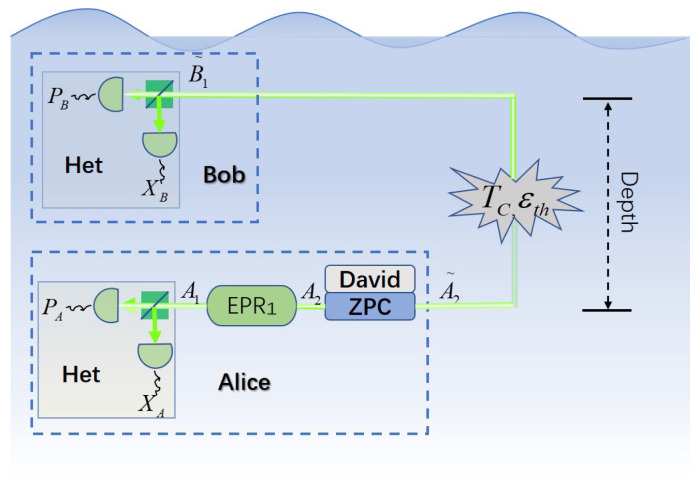
Schematic diagram of the ZPC-based point-to-point (PP) CVQKD system.

**Figure 3 entropy-22-00571-f003:**
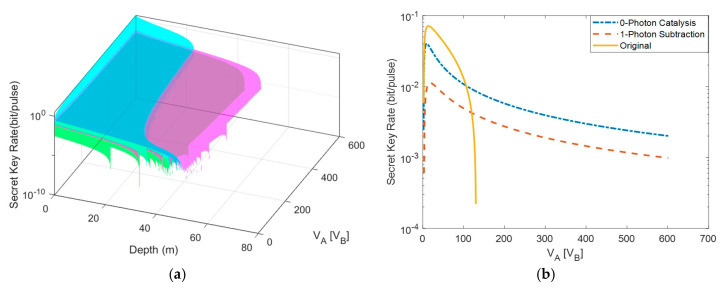
(**a**) The secret key rate as a function of $V_{A}$ ($V_{B}$) for the traditional scheme (blue surface) and the ZPC-based (magenta surface) and the single photon subtraction (SPS)-based scheme (green surface). (**b**) A cross section of (a) where depth is set to 30 m for the traditional (yellow), the ZPC-based (blue), and the SPS-based (red).

**Figure 4 entropy-22-00571-f004:**
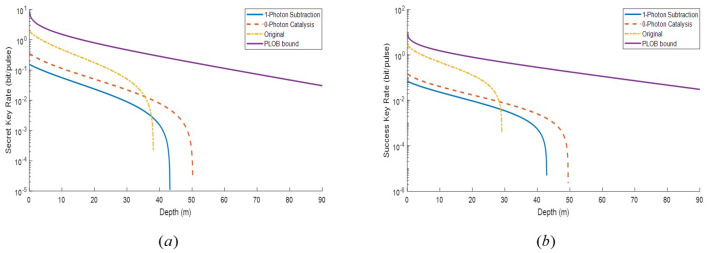
The secret key rate of the MDI-CVQKD system under pure seawater via the ZPC-based scheme, the SPS-based scheme, and the traditional scheme. *T*(SPS) = 0.9. The purple line represents PLOB [28] bound. (**a**). $V_{A}=V_{B}=40$. (**b**). $V_{A}=V_{B}=150.$

**Figure 5 entropy-22-00571-f005:**
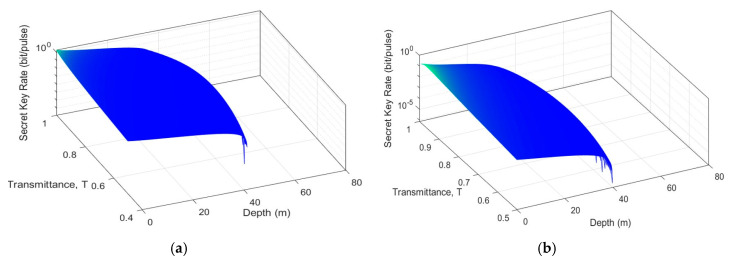
The secret key rate of the MDI-CVQKD system under pure seawater for $V_{A}=V_{B}=40$. (**a**) the ZPC-based scheme, (**b**) the SPS-based scheme.

**Figure 6 entropy-22-00571-f006:**
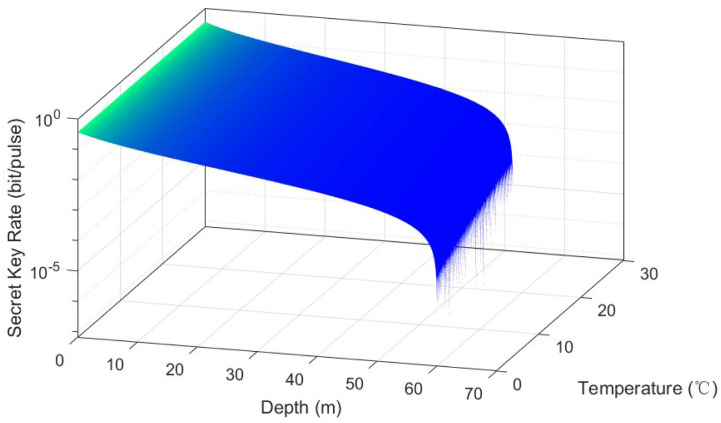
Relationship among secret key rate, transmission depth and temperature for $V_{A}=V_{B}=40$.

**Figure 7 entropy-22-00571-f007:**
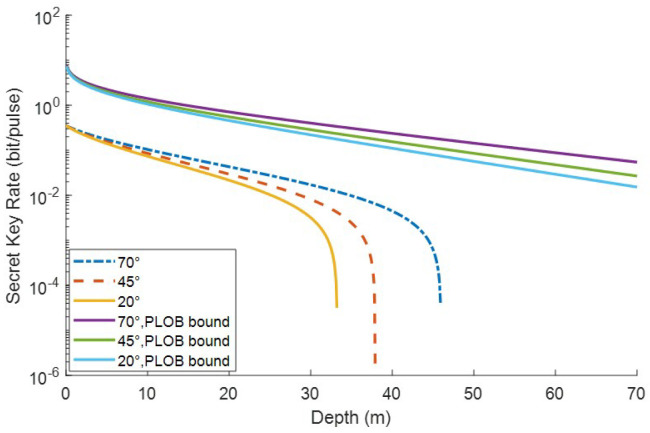
Secret key rate of the ZPC-based MDI-CVQKD in oligotrophic seawater under different sun elevation angle for $V_{A}=V_{B}=40$. The upper three lines represent PLOB bound corresponding different sun elevation angle.

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
