# Peer review of "Improving Underwater Continuous-Variable Measurement-Device-Independent Quantum Key Distribution via Zero-Photon Catalysis"

_entropy, 2020, doi:10.3390/e22050571_

Round 1

Reviewer 1 Report

Please see the enclosed file

Author Response

Dear Reviewer,

Thank you very much for your careful and constructive advices. Those comments are all valuable and very helpful for revising and improving our paper, and with the important guiding significance to our researches. We have tried our best to revise and improve the manuscript and made some changes in the manuscript.

We note that we have provided a point-by-point response to the reviewer’s comments as a Word/PDF file in the attachment. Please see the attachment.

Thank you again.

With best regards,
Yours sincerely,

Wang Yuang, Zou Shanhua, Mao Yun, Guo Ying

Reviewer 2 Report

I have read the paper “Improving underwater continuous-variable measurement-device-independent quantum key distribution via zero-photon catalysis” with great interest. I think that the authors did a good job but I am afraid that the manuscript as is now needs revisions to include relevant literature clearly missed by the authors.

It is surprising that the authors cite 9,10 for CV MDI QKD completely missing the actual paper which introduced this idea (as one can verify from the chronology on the arxiv). This is: https://arxiv.org/abs/1312.4104

published as Nature Photonics 9, 397-402 (2015).

This paper should be cited in the place of 9,10 or before 9,10.

Furthermore a citation to MDI is missing, which was co-proposed in the two PRLs:

Physical review letters 108 (13), 130502 (2012) - Braunstein, Pirandola

Physical review letters 108 (13), 130503 (2012) - Lo, Curty and Qi

The authors cite quite old reviews in the field. At the moment there is a large and updated review on quantum cryptography which is better than ref. 1.

My main technical point about the paper is how the rates found by the authors compare to the PLOB bounds for pure loss channel and thermal loss channel. Can they make a clear comparison, e.g., in their figs. 4,7 ? This would be an important sanity check in my opinion.

Note that in the figures that authors use “success key rate” instead of “secret key rate”.

Once these points have been addressed I am happy to provide a final recommendation.

Author Response

(The authors gave the same response as above.)

Round 2

Reviewer 1 Report

This is my second report. The manuscript shows some improvement. Using a
different set of parameters, the workable distance increases from about 30m to
50m. The authors also clarifies that Alice and Bob are moving targets so that
a fiber link between them is not feasible. They also correctly point out that
,as far as visible light is concern, there is no hope of drastically
increase the secure distance. So, there is only one question reminding,
namely, is this work good enough for publication in Entropy given that it has
improved the workable distance though not necessarily the key rate for short
distance under a limitation of the rather high opacity sea water channel? I
am sorry to say that my judgment is still negative. If Alice and Bob are
moving targets, it is much simpler for them to move close to the water
surface and communicate through QKD in air via photons in visible wavelength
with the help of periscopes --- the ones that sublimes are using all the time.
Then they can communicate with a much longer distance, of order of several
km with ease. If needed, they may even communicate via satellite link. Then,
it is not difficult, at least in theory, to increase the distance to order of
1000km. Since there is an obvious alternative that is much more effective, I
cannot recommend acceptance of this manuscript.

Reviewer 2 Report

I have read the new version which is improved. There are some residual points for the authors to address related to literature.

1) PLOB bound has been introduced as benchmark. However the relevant reference is missing. 
https://www.nature.com/articles/ncomms15043/

2) Scarani’s Review on QKD is outdated. The new review https://arxiv.org/abs/1906.01645 is a much better choice and should be considered as standard for the field.

After these final corrections, I am happy to provide my final recommendation.
